# Molecular Signatures Related to Inflammation and Angiogenesis in Patients with Lower Extremity Artery Disease, Abdominal Aortic Aneurysm, and Varicose Veins: Shared and Distinct Pathways

**DOI:** 10.3390/ijms26188786

**Published:** 2025-09-09

**Authors:** Daniel Zalewski, Paulina Chmiel, Przemysław Kołodziej, Marcin Feldo, Andrzej Stępniewski, Marta Ziaja-Sołtys, Joanna Łuszczak, Agata Stanek, Janusz Kocki, Anna Bogucka-Kocka

**Affiliations:** 1Chair and Department of Biology and Genetics, Medical University of Lublin, 4a Chodźki St., 20-093 Lublin, Poland; 2Randox Laboratories Ltd., Poznańska St., 00-680 Warszawa, Poland; 3Chair and Department of Vascular Surgery and Angiology, Medical University of Lublin, 11 Staszica St., 20-081 Lublin, Poland; 4ECOTECH-COMPLEX Analytical and Programme Centre for Advanced Environmentally Friendly Technologies, University of Marie Curie-Skłodowska, 39 Głęboka St., 20-612 Lublin, Poland; 5Department of Internal Medicine, Metabolic Diseases and Angiology, Faculty of Health Sciences in Katowice, Medical University of Silesia, Ziołowa 45/47 St., 40-635 Katowice, Poland; 6Department of Clinical Genetics, Chair of Medical Genetics, Medical University of Lublin, 11 Radziwiłłowska St., 20-080 Lublin, Poland

**Keywords:** angiogenesis, inflammation, lower extremity artery disease, abdominal aortic aneurysm, varicose veins, gene expression, plasma proteins, molecular signatures

## Abstract

Lower extremity artery disease (LEAD), abdominal aortic aneurysm (AAA), and varicose veins (VV) are frequently underdiagnosed and undertreated peripheral vascular diseases that pose considerable public health challenges. More research is required to elucidate the pathophysiological mechanisms underlying these conditions and to identify novel diagnostic and therapeutic biomarkers. Therefore, in our study, we aimed to identify shared and distinct pathways associated with angiogenesis and inflammation in LEAD, AAA, and VV. The expression of 18 genes in peripheral blood mononuclear cells and the plasma levels of six proteins were compared between groups of 40 patients with LEAD, 40 patients with AAA, and 40 patients with VV. Independent RNA-seq and microRNA-seq data were integrated to predict differentially expressed transcription factors and microRNAs associated with the most significant genes. Gene Ontology functional analysis was performed to determine the potential biological effects of the observed dysregulations. The elevated expression of *VEGFB* and *TGFB1*, along with increased plasma levels of VEGF-C and reduced plasma levels of VEGF-A, were distinguishing features of patients with LEAD compared to those with AAA and VV. Decreased plasma levels of TGF-alpha and TGF-beta 1 were found to be indicative of varicose veins compared to individuals with arterial diseases (LEAD and AAA). Transcription factors and microRNAs potentially regulating the obtained signatures were identified and integrated into a hypothetical regulatory network. The observed dysregulations were found to be functionally associated with the response to hypoxia, the positive regulation of angiogenesis, chemotaxis, vascular permeability, and cell adhesion. The presented study identified dysregulations of key angiogenesis- and inflammation-related factors in peripheral blood mononuclear cells and plasma between LEAD, AAA, and VV patients, providing new insights into the shared and distinct molecular mechanisms underlying these diseases.

## 1. Introduction

Vascular disorders represent a significant global medical challenge. Among these, diseases associated with cardiac circulation account for the highest mortality rates, prompting scientific research to predominantly focus on these conditions. Consequently, comparatively less attention has been paid to the etiopathogenesis, diagnostic strategies, and therapeutic approaches for diseases involving the peripheral vasculature (outside the coronary and cerebral circulation) [1,2,3].

Peripheral vascular diseases comprise a diverse group of conditions that includes, among others, atherosclerotic disorders (e.g., lower extremity artery disease), venous pathologies (e.g., varicose veins), and aneurysmal conditions (e.g., abdominal aortic aneurysm). Lower extremity artery disease (LEAD) is a progressive and multifactorial condition resulting from the formation of atherosclerotic plaques in the arteries of the lower limbs. This disease encompasses a range of pathophysiological mechanisms including endothelial dysfunction, lipid accumulation, hypoxic responses, chronic inflammation, angiogenesis, and vascular calcification. LEAD is estimated to affect approximately 230 million adults worldwide. It is strongly associated with an elevated risk of adverse clinical outcomes such as chronic limb-threatening ischemia and limb amputation, thereby significantly contributing to the global burden of atherosclerotic disease [1,4,5,6,7,8].

Abdominal aortic aneurysm (AAA) is a life-threatening condition characterized by focal weakening and dilation of the aortic wall. A severe complication is aneurysm rupture, which is associated with a mortality rate of up to 85%. The molecular mechanisms underlying the structural and functional changes in the abdominal aorta wall are linked to the dysregulation of key biological processes, including extracellular matrix degradation, inflammation, angiogenesis, apoptosis of vascular smooth muscle cells, endothelial dysfunction, and responses to oxidative stress. Because of its largely asymptomatic nature, AAA is often incidentally identified during imaging studies such as ultrasonography, computed tomography, or magnetic resonance imaging [9,10,11].

Varicose veins (VV) are the most prevalent manifestation of chronic venous disease and are characterized by abnormally dilated and tortuous superficial veins that predominantly affect the lower extremities. The primary etiological factors include venous hypertension, venous reflux, and microcirculatory dysfunction. These factors arise from hemodynamic disturbances in blood flow such as venous valve incompetence, venous obstruction, and dysfunction of the muscle pump [12,13,14]. If left untreated or inadequately managed, VV can lead to severe complications including venous ulcers and venous thromboembolic events [15,16,17,18].

Despite advancements in medical research and treatment, LEAD, AAA, and VV are frequently underdiagnosed and undertreated, and remain significant contributors to impaired quality of life, severe morbidity, and mortality. Given their high prevalence, multifactorial etiopathogenesis, and potential for severe complications, these diseases constitute significant global medical challenges [1,8,9,19,20]. Addressing these conditions requires increased attention and research efforts to elucidate the underlying pathophysiological mechanisms and identify novel diagnostic biomarkers.

Although LEAD, AAA, and VV are distinct vascular disorders, each characterized by unique clinical manifestations, outcomes, and treatment approaches, they share overlapping risk factors and common vascular pathophysiological processes, which can complicate their accurate diagnosis and management. Recent studies have suggested that an enhanced inflammatory response and dysregulated angiogenesis are significant contributors to the initiation and progression of these diseases. In LEAD, elevated levels of vascular endothelial growth factor A165b (VEGF-A165b) in muscle biopsy samples have been shown to inhibit endothelial VEGF receptors, thereby directly impairing angiogenesis, ischemic muscle revascularization, and perfusion recovery [21,22,23,24,25]. Additionally, circulating inflammatory cytokines, including interleukin-6, interleukin-8, C-reactive protein, and tumor necrosis factor-α, have emerged as indicators of LEAD [26,27]. In AAA, studies have indicated that heightened interleukin-8 signaling may contribute to mural inflammation and increased infiltration of T-helper lymphocytes into the aneurysmal wall [28,29,30]. Additionally, elevated plasma VEGF-A levels have been consistently associated with aneurysmal diseases, suggesting a role in the pathophysiological mechanisms underlying these conditions [30,31,32]. In VV, chronic damage to the vessel wall caused by turbulent blood flow is associated with enhanced inflammatory response. This is reflected by elevated circulating levels of intercellular adhesion molecule-1, vascular cell adhesion molecule-1, angiotensin-converting enzyme, L-selectin, and interleukin-6 [33,34]. Furthermore, varicose vein tissue exhibits increased expression of factors involved in the regulation of angiogenesis, including VEGF-A, VEGF receptor 2 (VEGFR-2), hypoxia-inducible factor-1α (HIF-1α), and metallothionein [35,36,37].

Despite increasing evidence indicating aberrant inflammation and angiogenesis in patients with LEAD, AAA, and VV compared to healthy controls, significant gaps remain in our understanding of the disease-specific differences in the signaling pathways regulating these processes. To address this gap, our research team focused on identifying distinctive and shared changes in the expression of key regulators of angiogenesis and inflammation among these conditions. Using previously acquired data, we analyzed the expression levels of 18 genes (*ANGPT1*, *ANGPT2*, *CCL2*, *CCL5*, *CSF2*, *CXCL8*, *FGF2*, *IL1A*, *IL1B*, *IL6*, *PDGFA*, *PDGFB*, *TGFA*, *TGFB1*, *TNF*, *VEGFA*, *VEGFB*, and *VEGFC*) in peripheral blood mononuclear cell (PBMC) samples, as well as the plasma concentrations of six proteins (ANGPT-1, ANGPT-2, TGF-alpha, TGF-beta 1, VEGF-A, and VEGF-C) in a cohort of 40 patients with LEAD, 40 with AAA, and 40 with VV. This dataset, which was previously employed to identify potential biomarkers for AAA and VV in comparison to healthy controls [30,37], was leveraged in the current study to directly compare the diseases. Furthermore, analysis of whole transcriptome RNA-seq and miRNA-seq expression datasets was integrated to identify transcription factors (TFs) and microRNAs (miRNAs), respectively, which may be involved in the dysregulation of genes observed between the studied diseases. This comprehensive approach aims to elucidate both the shared and distinct molecular features of the studied diseases at the levels of miRNAs, TFs, effector genes, and proteins (Figure 1).

## 2. Results

### 2.1. Characteristics of the Study Groups

The main clinical characteristics of patients in the LEAD (n = 40), AAA (n = 40), and VV (n = 40) groups in the gene expression and plasma protein datasets were compared across groups (Table 1). As expected, several differences reflecting the associations between risk factors and the diseases studied were observed. The LEAD group exhibited a significantly higher proportion of individuals with smoking habits, hypertension, and higher fibrinogen levels, all of which are strongly associated with LEAD. Additionally, the VV group comprised younger individuals with lower levels of low-density lipoprotein (LDL), creatinine, urea, and C-reactive protein (CRP) than the LEAD and AAA groups. The AAA group displayed markedly higher homocysteine levels compared to both the LEAD and VV groups. No statistically significant differences were observed between the groups in terms of sex, body mass index (BMI), or blood HDL and cholesterol levels.

### 2.2. Dysregulatios of Genes Related to Angiogenesis and Inflammation Between LEAD, AAA, and VV

The expression levels of 18 genes associated with angiogenesis and inflammation (*ANGPT1*, *ANGPT2*, *CCL2*, *CCL5*, *CSF2*, *CXCL8*, *FGF2*, *IL1A*, *IL1B*, *IL6*, *PDGFA*, *PDGFB*, *TGFA*, *TGFB1*, *TNF*, *VEGFA*, *VEGFB*, and *VEGFC*) were analyzed in the following comparisons: LEAD vs. AAA, LEAD vs. VV, and AAA vs. VV. During the data quality control process, four of the 18 analyzed genes (*ANGPT2*, *CSF2*, *IL1A*, and *IL6*), along with three samples, were excluded from the dataset due to low quality of data (Appendix A). Additionally, five samples displayed outlier expression patterns (Appendix A) and were consequently omitted from further analysis.

Differences in the expression levels of the 14 retained genes between the compared groups were represented as follows: fold change values calculated using the delta Ct method for relative quantification, area under the receiver operating characteristic curve (ROC-AUC) values derived from the receiver operating characteristic (ROC) method, and odds ratio (OR) values determined through univariate logistic regression. Genes with statistically significant fold change values (false discovery rate, FDR < 0.05) were considered as gene signatures (Table 2). The entire results are provided in Appendix A. The distributions of the normalized expression levels of all the analyzed genes (2^−dCt^ values) across the studied groups are presented in Figure 2.

Three genes (*TGFB1*, *VEGFA*, and *VEGFB*) were found to have significantly higher expression levels in the LEAD group than in the AAA group. In the comparison between the LEAD and VV groups, two genes (*TGFB1* and *VEGFB*) had significantly higher expression levels, whereas the expression levels of one gene (*CCL5*) were lower (Table 2). Comparison between the AAA and VV groups identified no genes that exhibited statistically significant differences in expression (Appendix A).

The genes identified as statistically significant consistently exhibited the highest ROC-AUC values in ROC analysis (Table 2, Appendix A). Among these, *VEGFB* demonstrated particularly high ROC-AUC values, underscoring its robust discriminatory capability to distinguish the studied LEAD subjects from those with AAA (ROC-AUC = 0.953) and VV (ROC-AUC = 0.876).

The differential character of the studied genes was further evaluated using logistic regression conducted for the same comparisons (LEAD vs. AAA, LEAD vs. VV, and AAA vs. VV) in both univariate and multivariate modes. Odds ratios (ORs) and associated FDR values were calculated to quantify the likelihood of the condition occurring when the mean of 2^−dCt^ values in the studied samples will double. Consistent with the findings from previous methods, genes exhibiting statistically significant changes in expression also displayed statistically significant ORs in the univariate logistic regression model (except for *CCL5*) (Table 2 and Appendix A).

In the multivariate logistic regression model, sex, age, BMI, and smoking status were included as covariates to determine whether differences in gene expression between the groups could be considered independent of these variables. The differential expression of only *TGFB1*, *VEGFA*, and *VEGFB* between the LEAD and AAA groups remained statistically significant after adjustment for these variables, suggesting that these genes may be considered independent (Appendix A). Conversely, gene signatures from LEAD vs. VV comparison (*CCL5*, *TGFB1*, and *VEGFB*) lost statistical significance; therefore, the expression of these genes might be associated with other variables. Therefore, an analysis of such potential associations was conducted, and the results are detailed in Section 2.4.

The final approach employed to evaluate the diagnostic significance of differential expression of selected gene signatures (*CCL5*, *TGFB1*, *VEGFA*, and *VEGFB*) was a decision tree model created using the C5.0 method (Appendix A). The model demonstrated an overall classification accuracy of 81.25%, with the highest importance of *TGFB1*. This model achieved high classification accuracies of 100% for the LEAD group and 92.5% for the AAA group, while the classification performance was comparatively low for patients with VV (48.57% accuracy).

The results suggest that increased expression of *TGFB1*, *VEGFA*, and *VEGFB* in PBMC can effectively distinguish LEAD patients from those with AAA. Additionally, the increased expression of *TGFB1* and *VEGFB*, together with the reduced expression of *CCL5* in PBMC can differentiate LEAD patients from VV subjects. These observed dysregulations likely reflect differences in the angiogenic and inflammatory pathways between these conditions; however, more research is required to validate and further elucidate the clinical relevance of these findings.

### 2.3. Differences in Plasma Concentrations of Angiogenesis-Related Proteins in LEAD, AAA, and VV

The plasma concentrations of six key regulators of angiogenesis (ANGPT-1, ANGPT-2, TGF-alpha, TGF-beta 1, VEGF-A, and VEGF-C) were compared among the LEAD, AAA, and VV groups, employing the same comparative framework used for the gene expression analysis (LEAD vs. AAA, LEAD vs. VV, and AAA vs. VV). The number of samples in which the concentrations of the analyzed proteins were either quantifiable or fell below the detection range of the used ELISA kits is summarized in Appendix A. Among the six proteins examined, only ANGPT-1, ANGPT-2, and TGF-beta 1 exhibited concentrations within the quantification range in the majority of samples across all groups. TGF-alpha levels were predominantly below the detection threshold in the LEAD and AAA groups, whereas VEGF-C levels were frequently below the detection range in the AAA and VV groups. Additionally, VEGF-A concentrations fell below the detection threshold in the majority of the samples within the LEAD group.

Hierarchical clustering and principal component analysis (PCA) identified a clear outlier within the dataset (Appendix A, respectively). The outlier status of this sample was primarily attributed to an exceptionally elevated VEGF-C level, suggesting the possibility of distinct underlying conditions in the corresponding patient. Consequently, this sample was excluded from further analysis

Statistically significant differences in plasma protein levels between the compared groups (FDR < 0.05) are presented in Table 3. The results for all analyzed proteins are presented in Appendix A. The extended results of the ROC analysis are presented in Appendix A. The distribution of the plasma levels for the six analyzed proteins across the study groups is illustrated in Figure 3.

The mean ANGPT-1 concentration was significantly higher in the VV group than in the LEAD group (ROC-AUC = 0.764). Similarly, the mean concentrations of both TGF-alpha and TGF-beta 1 were significantly elevated in the VV group relative to those in the LEAD and AAA groups (ROC-AUC > 0.7). Furthermore, subjects with LEAD exhibited significantly lower plasma levels of VEGF-A and significantly higher plasma levels of VEGF-C compared to the other groups (ROC-AUC > 0.7). No statistically significant differences in plasma levels of ANGPT-2 were observed among the study groups.

Similarly to the gene expression analysis, the differential character of the studied proteins was further assessed using logistic regression applied to the same comparisons (LEAD vs. AAA, LEAD vs. VV, and AAA vs. VV) in both univariate and multivariate frameworks. Odds ratios and corresponding FDR values were computed to quantify the probability of the condition occurring in response to a doubling of the mean plasma concentrations in the analyzed samples. Consistent with the results obtained using earlier methods, proteins demonstrating statistically significant changes in expression also exhibited significant ORs in the univariate logistic regression model (Table 3 and Appendix A).

The multivariate logistic regression model incorporated the same covariates: sex, age, BMI, and smoking status. Plasma concentrations of VEGF-A and VEGF-C between the LEAD and AAA groups remained statistically significant after adjusting for these variables, indicating that the observed differences in plasma levels of these proteins between the groups are likely independent of the included covariates (Appendix A). In turn, proteins that lost statistical significance may be associated with other variables. An additional analysis was performed to investigate these potential associations, with the results presented in Section 2.4.

The decision tree method was used to assess the diagnostic performance of five selected proteins (ANGPT-1, TGF-alpha, TGF-beta 1, VEGF-A, and VEGF-C), yielding an overall classification accuracy of 90.76% (Appendix A). Group-specific stratification revealed a high classification accuracy for the LEAD and AAA groups (95%), whereas in the VV group, the accuracy was comparatively lower (82.05%). The second approach involves the evaluation of a model that integrates the expression profiles of four selected genes (*CCL5*, *TGFB1*, *VEGFA*, and *VEGFB*), along with the plasma concentrations of five selected proteins. This model demonstrated a superior overall accuracy of 99.1%, misclassifying only a single sample from the VV group as belonging to the LEAD group (Appendix A).

Intriguingly, a higher expression of *TGFB1* but lower plasma levels of TGF-beta 1 were found in the LEAD group than in the VV group. Similarly, higher expression of *VEGFA*, but lower plasma levels of VEGF-A were found in the LEAD group compared to the AAA group. This suggests that the plasma levels of these factors do not reflect their expression in PBMC of patients with these diseases.

The findings obtained from gene expression and plasma protein level analyses suggest that higher expression of *VEGFB* and *TGFB1*, higher plasma levels of VEGF-C, and lower plasma levels of VEGF-A are distinguishing characteristics of patients with LEAD, differentiating them from those with AAA and VV. Furthermore, lower plasma levels of TGF-alpha and TGF-beta 1 may indicate venous pathology associated with varicose veins, and could distinguish individuals with arterial diseases (LEAD and AAA) (Figure 4). Nonetheless, further investigation is needed to validate these findings and to assess the clinical utility of these protein biomarkers in differentiating LEAD, AAA, and VV.

### 2.4. Relationships Between Identified Molecular Signatures and Characteristics of the Study Groups

Analysis of four genes (*CCL5*, *TGFB1*, *VEGFA*, and *VEGFB*) and five proteins (ANGPT-1, TGF-alpha, TGF-beta 1, VEGF-A, and VEGF-C), which demonstrated statistically significant differences in the performed comparisons, was expanded to investigate potential associations with the risk factors and biochemical parameters of the study population. Correlation analysis and univariate linear regression were employed to evaluate the relationships with continuous variables (age, BMI, and blood levels of cholesterol, LDL, HDL, homocysteine, urea, fibrinogen, creatinine, and C-reactive protein). These analyses were performed across all groups (general mode) and within individual groups (group-specific mode). For categorical clinical variables (sex, smoking, and hypertension), statistical comparisons were conducted using the Mann–Whitney U test or Student’s *t*-test depending on the normality of the data distribution. Due to the limited balance of smoking and hypertension among the groups, analysis of these characteristics was restricted to the general mode (across all groups).

Correlation analysis conducted for all groups identified only two associations with correlation coefficients (R) exceeding 0.4: one between the expression of *VEGFB* and fibrinogen blood levels (R = 0.55, FDR = 9.423 × 10^−9^), and another between TGF-alpha plasma concentrations and C-reactive protein blood levels (R = −0.45, FDR = 1.310 × 10^−5^). In the group-specific analysis, significant correlations (R > 0.4, FDR < 0.05) were observed in the VV group between blood urea levels and the expression of *CCL5*, *TGFB1*, *VEGFA*, and *VEGFB* (Appendix A). These relationships were further confirmed to be significant (*p* < 0.05) in the univariate linear regression analysis.

In the analysis of categorical clinical variables, no statistically significant differences were identified in the selected molecular signatures between the male and female subjects. However, significant variations were observed between smokers and non-smokers in the expression levels of *CCL5* and *VEGFB*, and the plasma concentrations of ANGPT-1, TGF-alpha, VEGF-A, and VEGF-C. Furthermore, these genes and proteins, along with *TGFB1* and *VEGFA*, demonstrated significant differences between individuals with and without hypertension (Appendix A).

### 2.5. Coexpression of Selected Genes and Proteins

To investigate the similarity in the expression patterns of the studied genes and proteins, a pairwise correlation analysis was performed for four genes (*CCL5*, *TGFB1*, *VEGFA*, and *VEGFB*) and five proteins (ANGPT-1, TGF-alpha, TGF-beta 1, VEGF-A, and VEGF-C) selected as statistically significantly differentiating the compared groups. Similarly to the previous correlation analysis, either analysis across all groups (general mode) or within individual groups (group-specific mode) was performed (Appendix A). The strongest correlations (R > |0.6|) are summarized in Figure 5.

The findings indicate that *TGFB1* is co-expressed with *VEGFA* and *VEGFB* across all studied diseases, which likely reflects the regulatory and functional associations between these genes. The correlation between *TGFB1* and *VEGFA* was most pronounced in patients with LEAD, whereas the correlation between *TGFB1* and *VEGFB* was predominantly driven by the AAA group. Similar correlations were observed in the VV group; however, this group exhibited additional mutual correlations between *VEGFA*, *VEGFB*, and *CCL5*. These interactions may represent mechanisms specific to venous vascular pathology; however, further studies are required to validate this conclusion.

### 2.6. Identification of Transcription Factors (TFs) Potentially Involved in the Observed Changes in Gene Expression

To further investigate the regulatory mechanisms that potentially contribute to the altered expression of the genes selected from the comparative analyses, TFs potentially associated with these genes were identified. TFs regulating these genes were identified using the TFLink database [38], yielding 158, 481, 615, and 337 TFs for *CCL5*, *TGFB1*, *VEGFA*, and *VEGFB*, respectively (Appendix A).

The expression levels of these TFs were compared between independent LEAD and AAA, as well as LEAD and VV groups, using the DESeq2 package [39] and the previously obtained RNA-seq dataset. Quality control analysis verified the consistency of the normalized data, confirming the absence of outlier samples (Appendix A) and high quality of the differential expression results (Appendix A).

Five differentially expressed TFs (*EBF1*, *NFIL3*, *PAX5*, *PPARG*, and *SOX5*) met the significance criteria (FDR < 0.05, |log_2_ fold change| > 1, and a mean normalized count > 10) and were selected as potentially functionally associated with at least one of the four genes selected from corresponding comparisons (Appendix A). The differential expression of selected TFs was further evaluated using ROC analysis (Appendix A) and visualized on boxplots (Appendix A).

*PPARG* was identified as being significantly differentially expressed in the LEAD vs. AAA comparison and was associated with the transcriptional regulation of *TGFB1*, *VEGFA*, and *VEGFB*. Both *PPARG* and its associated target genes were upregulated in patients with LEAD compared with those with AAA. The likelihood of these positive regulatory interactions is further supported by the strong co-expression of *TGFB1*, *VEGFA*, and *VEGFB* (Figure 3), suggesting a shared regulatory mechanism.

Four other TFs, *EBF1*, *NFIL3*, *PAX5*, and *SOX5* were selected from the LEAD versus VV comparison. *NFIL3* exhibited higher expression in LEAD patients in relation to those with VV, and was associated with *VEGFB*, which was also upregulated in this comparison. In turn, *PAX5* was downregulated in the LEAD vs. VV comparison and linked to *CCL5*, which was also downregulated in this comparison. *EBF1* and *SOX5* exhibited inverse expression patterns relative to their corresponding regulated genes (Table 4), suggesting that these transcription factors may exert suppressive regulatory effects on their target genes.

The interactions between these TF–target gene pairs could be involved in the molecular mechanisms underlying the pathophysiological differences observed between LEAD and VV.

### 2.7. Identification of miRNA Potentially Involved in the Observed Changes in Expression of Selected Genes and TFs

The next phase of this study aimed to investigate the potential involvement of miRNA-mediated regulatory mechanisms in the altered expression of four selected genes (*CCL5*, *TGFB1*, *VEGFA*, and *VEGFB*) and associated TFs (*EBF1*, *NFIL3*, *PAX5*, *PPARG*, and *SOX5*). Experimentally validated miRNAs targeting these genes and TFs were identified using the miRNet database (https://www.mirnet.ca/) [40]. This analysis yielded a set of 577 unique miRNAs that potentially interact with these genes and TFs (Appendix A).

To determine which of these miRNAs may contribute to the dysregulation of the analyzed genes and TFs, the subset of 577 obtained miRNAs was extracted from previously published miRNA-seq dataset [41,42,43]. Expression data regarding these miRNAs were subjected to differential expression analysis (LEAD vs. AAA and LEAD vs. VV) performed using the DESeq2 package in R [39]. Quality control analysis verified the consistency of the normalized data, confirming the absence of outlier samples (Appendix A) and high quality of the differential expression results (Appendix A). Based on the assumption that miRNAs primarily suppress the expression of their target genes, our approach aimed to identify upregulated miRNAs associated with downregulated genes or TFs, and vice versa. Seventeen unique miRNAs that satisfied this assumption and met the established statistical significance criteria (FDR < 0.05, |log_2_ fold change| > 0.3, and a mean of normalized count > 10), were selected as potential regulators of the analyzed genes and TFs (Table 5 and Appendix A). The differential expression of the selected miRNAs was confirmed using distribution comparisons (Appendix A) and ROC analysis (Appendix A).

Results obtained from this analysis was integrated with the results of real-time PCR and RNA-seq results, providing a valuable framework for construction a predictive regulatory network that integrates selected genes, TFs, and miRNAs (Figure 6). In the LEAD vs. AAA comparison, two downregulated miRNAs (hsa-miR-1301-3p and hsa-miR-326) were identified as potentially targeting of upregulated *VEGFA*, while one downregulated miRNA (hsa-miR-491-5p) was found to presumably interact with the upregulated *PPARG*. In the LEAD vs. VV comparison, one downregulated miRNA (hsa-miR-146a-5p) was likely associated with the upregulation of *TGFB1*. Four downregulated miRNAs (hsa-miR-1229-3p, hsa-miR-625-3p, hsa-miR-335-5p, and hsa-miR-491-5p) were in the group of miRNAs that presumably interact with the upregulated *NFIL3*. Interestingly, hsa-miR-491-5p exhibited reduced expression in both comparisons and was predicted to target both *PPARG* and *NFIL3*, which may suggest a potential regulatory mechanism underlying the increased expression of these genes (Figure 6).

Furthermore, in the LEAD vs. VV comparison, several upregulated miRNAs were predicted to target the downregulated *CCL5* gene and three TFs (*EBF1*, *PAX5*, and *SOX5*). Interestingly, hsa-miR-34a-5p was indicated to target all these genes, further supporting its potential role in the transcriptional regulation that differentiates LEAD from VV (Figure 6).

The identified miRNA-TF-effector gene interactions may represent key regulatory mechanisms that influence the differential expression of genes involved in angiogenesis and inflammation, contributing to the distinct molecular profiles observed in patients with LEAD, AAA, and VV. However, the constructed network has hypothetical character; therefore, further studies are required to validate these findings.

### 2.8. Identification of Biological Processes Related to Identified Gene and Protein Signatures

Genes and proteins identified as signatures distinguishing the studied diseases are well-established regulators of inflammation and angiogenesis. However, a more comprehensive understanding of associated molecular processes is necessary. To achieve this, a functional analysis was conducted using overrepresentation analysis within the Panther 19.0 database (https://pantherdb.org/). This analysis was performed for gene and protein signatures identified as upregulated in the LEAD vs. AAA comparison (*TGFB1*, *VEGFA*, and *VEGFB*), upregulated in the LEAD vs. VV comparison (*VEGFB*, VEGF-C), downregulated in the LEAD vs. VV comparison (*CCL5*, ANGPT-1, TGF-alpha, VEGF-A), and downregulated in the AAA vs. VV comparison (TGF-alpha, TGF-beta 1). Gene Ontology (GO) Biological Process terms with a FDR < 0.05 were identified, and the most specific terms at the lowest hierarchical level (terminal descendants) were selected (Appendix A). A total of 35 unique terms were retrieved for the analyzed sets, except for the downregulated signatures in the AAA vs. VV comparison, for which no terms met the FDR < 0.05 threshold. The unique and shared terms across the analyzed factor sets are presented in Figure 7.

The GO terms uniquely associated with signatures distinguishing LEAD from AAA included response to hypoxia and positive regulation of angiogenesis, whereas those differentiating LEAD from VV were linked to the regulation of vascular permeability and cell adhesion. Additionally, the signatures differentiating LEAD from both AAA and VV were associated with positive regulation of chemotaxis and VEGF signaling, and these processes were implicated in both upregulated and downregulated signatures.

## 3. Discussion

In the present study, distinct and shared dysregulations of the main regulators of angiogenesis and inflammation were identified among patient groups with three peripheral vascular diseases: LEAD, AAA, and VV. Two primary datasets were used for the analysis. The first dataset contained the expression levels of 18 genes (*ANGPT1*, *ANGPT2*, *CCL2*, *CCL5*, *CSF2*, *CXCL8*, *FGF2*, *IL1A*, *IL1B*, *IL6*, *PDGFA*, *PDGFB*, *TGFA*, *TGFB1*, *TNF*, *VEGFA*, *VEGFB*, and *VEGFC*) obtained in PBMC samples of 40 patients with LEAD, 40 patients with AAA, and 40 patients with VV. The second dataset comprised the plasma levels of six proteins (ANGPT-1, ANGPT-2, TGF-alpha, TGF-beta 1, VEGF-A, and VEGF-C) measured using ELISA in the same patient groups. Comparative analyses of gene expression and plasma protein levels were conducted between the groups (LEAD vs. AAA, LEAD vs. VV, and AAA vs. VV) to identify signatures with the most significant differences. Furthermore, the integration of miRNA-seq and RNA-seq data allowed the prediction of miRNAs and TFs potentially responsible for the observed dysregulation of gene signatures (Figure 1).

Given that the altered course of angiogenesis and inflammation has been implicated in all of these diseases, the identification of some common dysregulations was anticipated. However, distinct molecular signatures have also been indicated. For example, significantly higher expression of *VEGFA* in PBMC was observed exclusively in patients with LEAD compared to those with AAA (Figure 4). VEGF-A is recognized as a key factor in the pathogenesis of atherosclerosis in LEAD, promoting leukocyte adhesion to endothelial layers, enhancing endothelial permeability, and facilitating transendothelial migration and leukocyte activation [44,45]. Interestingly, VEGF-A overexpression has also been reported in both PBMC and aortic wall of human and experimental AAA, where it plays a role in AAA-related processes such as neoangiogenesis, infiltration of inflammatory cells, MMP activity, and ECM degradation [30,31,46,47]. The increased *VEGFA* expression observed in PBMC from patients with LEAD compared to those with AAA in our study may indicate that VEGF-A-mediated pro-inflammatory effects are more pronounced in LEAD. This hypothesis is further supported by the results of the functional enrichment analysis of genes with higher expression in the LEAD group compared to AAA (and also VV), which revealed significant enrichment in the VEGF signaling pathway and the positive regulation of chemotaxis (Figure 7).

The influence of *VEGFA* on the course of angiogenesis in these diseases is challenging to assess, as different transcript variants of the *VEGFA* gene exert opposing effects on angiogenesis, depending on the form of exon 8 [48,49,50]. Studies suggest that the anti-angiogenic VEGF-A165b isoform contributes to the pathogenesis of LEAD by inhibiting endothelial VEGFR1 and VEGFR2 receptors, thus impairing angiogenesis, ischemic muscle revascularization, and perfusion recovery [21,22,23,24]. However, genes with higher expression in patients with LEAD vs. those with AAA were significantly associated with the positive regulation of angiogenesis (Figure 7). This contradiction underscores the complexity of the role of VEGF-A and highlights the need for further investigation of its involvement in LEAD and AAA. Moreover, the observed differences in *VEGFA* expression in PBMC and its plasma concentrations between patient groups with these diseases appeared to be divergent (Figure 2 and Figure 3), further emphasizing the necessity for comprehensive studies to clarify its contribution to disease pathogenesis.

In our study, elevated expression of *VEGFA* in PBMC was strongly correlated with the expression of *TGFB1* and *VEGFB* (Figure 5). Previous studies have shown that TGF-beta promotes the expression of VEGF [51], which may explain the observed co-expression of these genes. The increased expression of *TGFB1* in LEAD is probably driven by pro-fibrotic factors, such as platelet-derived growth factor (PDGF)-BB and connective tissue growth factor (CTGF). Increased secretion of these factors has been shown to increase in response to ischemia–reperfusion cycles [52], which are characteristic clinical hallmarks of LEAD.

Our study proposes a novel regulatory mechanism involving the transcription factor *PPARG*, which was predicted to be potentially responsible for the higher expression of *TGFB1*, *VEGFA*, and *VEGFB* in PBMC of LEAD patients compared to AAA subjects (Figure 6). Increased *PPARG* expression has previously been observed in atherosclerotic plaques in experimental models and has been associated with anti-inflammatory and anti-atherosclerotic effects [53,54,55]. Although suggested in our study as LEAD-specific, the regulatory crosstalk between *PPARG* and the VEGF and TGF-beta signaling pathways has been described in earlier studies [56,57], its precise regulatory role in atherosclerosis development requires further investigation.

Previous studies have identified several miRNA-regulatory axes involved in *PPARG* regulation in atherosclerosis models (e.g., miRNA-130a/PPARG/NF-κB and miRNA-19b/PPARG/NF-κB) [58,59]. Our study proposes an alternative miRNA-dependent regulatory mechanism of *PPARG* in LEAD, highlighting hsa-miR-491-5p as a potential regulatory factor (Figure 3). This mechanism is supported by a previous study in which *PPARG* was identified as a hub gene among the targets of hsa-miR-491-5p [60]. This regulatory mechanism may be specific to LEAD with less importance in AAA; however, another study reported the significance of PPARG protein in the proper production and integration of elastic fibers during AAA development [61].

Interestingly, a previous study demonstrated that a reduced expression of miR-491-5p activates the MT2-mediated HIF-1α/VEGF/MAPK pathway, thereby promoting angiogenesis in the injury site after traumatic brain injury in mice [62]. Observed in our study lower expression of this miRNA in LEAD patients vs. AAA and VV groups may suggest that a similar response may be triggered by vascular injury during atherosclerosis; however, further studies are necessary to validate this hypothesis.

Our study also shows that the higher expression of *TGFB1* and *VEGFB* observed in LEAD patients compared to VV patients may be driven by *NFIL3* (Figure 6). *NFIL3* has been implicated in the progression of atherosclerosis through the regulation of inflammatory responses, macrophage polarization, and lipid metabolism [63]. The pro-inflammatory effects of *NFIL3* have been associated with atherosclerosis in patients with rheumatoid arthritis [64]. *NFIL3* promotes endothelial cell injury by upregulating *ITGAM* transcription [65] as well as induce endothelial-mesenchymal transition by increasing the expression of the MEST gene [66]. These effects can be linked to the regulation of vascular permeability, which is associated with genes differentiating the LEAD and VV groups (Figure 7).

PBMC from patients with VV exhibited higher expression of *CCL5* and its transcription factors *EBF1* and *PAX5*, than those from patients with LEAD (Figure 6). CCL5, a member of the C-C motif chemokine family, is a key pro-inflammatory chemokine known to induce the migration and recruitment of immune cells such as T cells, dendritic cells, eosinophils, NK cells, mast cells, and basophils [67]. Higher expression levels of *CCL5* have previously been shown in PBMC and vein samples from VV cases than in normal samples [37,68]. Increased concentrations of *CCL5*-encoded protein have also been reported in blood from the varicose vein site compared to systemic concentrations [69]. *PAX5*, in turn, plays an important role in B-cell lineage commitment and maturation, and represents a potent oncogene in hematological cancers [70]. The second *CCL5*-related TF, *EBF1*, is implicated in blood pressure and inflammation, and its genetic variants were previously associated with varicose veins in a large-scale genetic study [71]. These findings may suggest that increased CCL5 signaling, potentially induced by *PAX5* and *EBF1*, may serve as a pro-inflammatory factor with higher significance in VV than in LEAD. Furthermore, our study showed that the increased expression of *CCL5*, *EBF1*, and *PAX5* may result from reduced expression of hsa-miR-34a-5p (Figure 6); however, in contrast, administration of miR-34a-3p and -5p in a circular dumbbell RNA form has been shown to induce *CCL5* expression in pancreatic cancer cells and suppress angiogenesis in HUVECs and zebrafish embryos [72]. Therefore, the role of the hsa-miR-34a-5p/CCL5 axis in VV needs to be validated in future studies.

The higher plasma levels of pro-angiogenic ANGPT-1 in VV group compared to LEAD patients (Figure 3) may suggest enhanced ANGPT-1-mediated pro-angiogenic and pro-inflammatory effects in VV. However, the role of ANGPT-1 in atherosclerosis has also been previously described [73,74].

Plasma levels of two TGF family factors, TGF-alpha and TGF-beta 1, were significantly higher in the VV group compared to LEAD and AAA groups (Figure 3). This difference was particularly pronounced for TGF-alpha, although its encoding gene exhibited similar expression in PBMC across all three diseases (Figure 2). Although studies focusing on the role of TGF-alpha in VV are limited, it has been shown to promote cellular proliferation and survival as well as modify the tumor microenvironment to facilitate vasculogenesis and metastasis [47,75]. Therefore, it can be hypothesized that the pro-proliferative effects of TGF-alpha may be associated with the vascular remodeling and hypertrophy observed in VV and potentially constitute a distinguishing feature of VV relative to LEAD and AAA.

In summary, our study identified distinct and shared gene and protein molecular signatures between patients with LEAD, AAA, and VV. By integrating RNA-seq and miRNA-seq data, we constructed predictive miRNA/TF/gene regulatory networks that may be involved in the dysregulation of inflammation and angiogenesis in these diseases (Figure 6). However, all regulatory associations were predicted based on the analysis of gene and protein expression data, and thus remain hypothetical and require validation in further mechanistic and functional studies.

The present study focused on circulatory markers, prompting the question of the extent to which alterations in these markers reflect the pathological mechanisms occurring within the vascular wall during disease development. Interactions between circulatory cells and pathologically altered vascular walls may induce the acquisition of novel cellular characteristics, potentially serving as indicators for disease detection. Furthermore, proteins and other biomolecules secreted from pathologically affected vascular sites may act as reliable biomarkers of disease progression. However, we observed that the expression patterns of genes in PBMC did not consistently correspond with the plasma protein levels of the same factors. Specifically, in contrast to elevated plasma concentrations of TGF-beta 1, the expression of the *TGFB1* gene in PBMC was significantly lower in the VV group compared with the LEAD group. A similar discordance was observed for VEGF-A: plasma levels were higher in the AAA group compared with the LEAD group, whereas *VEGFA* gene expression in PBMC was lower in this comparison (Figure 2 and Figure 3). These discrepancies complicate the interpretation of the role of these mediators in the vascular pathologies under study and suggest a potential tissue-specific regulation of their expression and secretion. Such inconsistencies between PBMC-derived transcriptomic data and plasma protein concentrations likely arise because plasma proteins originate from multiple tissue sources beyond PBMC. Moreover, PBMC mRNA expression does not necessarily translate into measurable protein production, and even when proteins are synthesized, they may remain intracellular rather than being secreted into the circulation. Furthermore, tissue-specific responses to pathological processes may give rise to distinct molecular signatures, reflecting the diverse mechanisms underlying disease pathogenesis and potentially leading to inconsistent results. Finally, these discrepancies may also stem from methodological differences, as various techniques can detect distinct isoforms or variants of the analyzed mRNAs and their corresponding proteins.

Previous studies have reported weak positive correlations between PBMC-derived *VEGF* expression and circulating VEGF levels in patients with acute myocardial infarction [76,77]. The data from our cohort revealed no significant association between VEGFA mRNA expression in PBMC and plasma VEGF-A concentrations (R = –0.09, *p* = 0.351). Similarly, no consistent relationship between PBMC expression and plasma levels of TGF-beta 1 has been demonstrated in patients with plaque morphea [78]. Our findings are concordant with these observations, as we also detected no significant correlation (R = –0.04, *p* = 0.715).

This lack of correlation can be explained by the fact that platelets represent the predominant source of circulating TGF-beta 1 [79], meaning that plasma concentrations primarily reflect platelet and tissue release rather than ongoing transcriptional activity within PBMC. Moreover, genetic studies indicate that under certain conditions, shared regulatory variants may influence both PBMC gene expression and plasma protein abundance, suggesting that concordance can occur, but is strongly disease- and context-dependent [80,81].

The present study has several limitations. The investigations conducted focused exclusively on selected key regulators of inflammation and angiogenesis; however, these processes are inherently complex and involve numerous receptors and downstream effectors, the analysis of which is beyond the scope of this study. Due to resource constraints, plasma protein levels were not measured for all factors analyzed at the gene expression level. The analyzed genes and proteins can form multiple transcript variants and isoforms, which may exert distinct, and in some cases even opposing, biological functions. The present study did not encompass a detailed assessment of these variants and isoforms, as it was intended to maintain a screening-oriented character. Therefore, future investigations should aim to perform an in-depth analysis of the significant factors identified in this study, with particular attention to their variant- and isoform-specific roles.

Additionally, it remains unclear whether the observed dysregulations are causative factors or a secondary hallmarks of the disease. The conclusions and hypotheses proposed in the discussion section are speculative and require further investigation in future studies.

While several genes and proteins exhibit promising diagnostic potential, several limitations should be considered when evaluating their clinical applicability. In our study, the sample size was limited, preventing the derivation of definitive conclusions regarding the observed associations. Furthermore, the heterogeneity of the studied diseases poses a significant challenge, as biomarker expression may vary according to disease stage, extent of vascular involvement, and presence of comorbid conditions such as diabetes, hypertension, or dyslipidemia. Therefore, translating biomarker research into routine clinical practice requires rigorous validation in large, prospective, and well-characterized patient populations, as well as cost-effectiveness analyses to ensure feasibility in real-world healthcare settings.

Nevertheless, despite these limitations, the results provide preliminary insights into the differences and similarities in the expression of key regulators of inflammation and angiogenesis among LEAD, AAA, and VV. Following validation in comprehensive studies, the identified dysregulations could potentially elucidate the underlying pathophysiological mechanisms, improve prognostic accuracy, and facilitate the identification of novel therapeutic targets.

## 4. Materials and Methods

### 4.1. Outline of the Study Design

The objective of this study was to identify molecular signatures that distinguish three diseases: LEAD, AAA, and VV. The workflow of this study is summarized in Figure 1.

This study utilized a gene expression and plasma protein datasets (refer to Data Availability Statement for access links) previously generated for our prior studies regarding analysis of 18 genes in PBMC and 6 plasma proteins associated with the regulation of angiogenesis and inflammation in patients with LEAD, AAA, and VV, compared to non-diseased control subjects [30,37] (the study concerning LEAD is currently undergoing the publication process in another peer-reviewed journal). In contrast, the current study adopted a different approach, aiming to compare the expression levels of the studied genes and proteins between the studied diseases. Comparative analyses were conducted between the LEAD and AAA groups, LEAD and VV groups, and AAA and VV groups. Genes and proteins exhibiting statistically significant differences between the groups were identified.

Furthermore, correlations between these differentially expressed genes and proteins, as well as associations with risk factors and biochemical parameters, were investigated. To elucidate the regulatory mechanisms underlying the altered expression of selected genes, associated TFs and miRNAs were identified, and their expression was compared across the independent LEAD, AAA, and VV groups using publicly accessible RNA-seq and miRNA-seq datasets (refer to Data Availability Statement for access links). A regulatory network comprising dysregulated genes, TFs, and miRNAs was subsequently constructed. Finally, functional enrichment analysis was performed to reveal the biological processes associated with the dysregulated genes and proteins.

Detailed descriptions of the methodologies used to generate all used datasets are available in previous studies [30,37,41,82,83] and for the readers’ convenience, also outlined in the Appendix A. The analysis of these datasets in the current study was conducted using R 4.3.2 programming software (https://www.r-project.org).

### 4.2. Gene Expression Dataset Analysis

This dataset contains information about the expression of 18 genes (*ANGPT1*, *ANGPT2*, *CCL2*, *CCL5*, *CSF2*, *CXCL8*, *FGF2*, *IL1A*, *IL1B*, *IL6*, *PDGFA*, *PDGFB*, *TGFA*, *TGFB1*, *TNF*, *VEGFA*, *VEGFB*, and *VEGFC*) in PBMC from 40 patients with LEAD (LEAD group), 40 patients with AAA (AAA group), and 40 patients with VV (VV group). Demographic and clinical characteristics of the study participants are shown in Table 1. Gene expression levels were determined using real-time PCR and specific TaqMan assays (Appendix A), and data analysis was conducted using the RQdeltaCT 1.3.2 package [84].

Low-quality Ct values, defined as exceeding 35, flagged as ‘undetermined’ or ‘inconclusive’, or having an AMPSCORE parameter of less than one, were identified for genes (Appendix A) and samples (Appendix A), and removed from the data. Four genes (*ANGPT2*, *CSF2*, *IL1A*, and *IL6*) were excluded from the study because of low-quality data in more than 50% of subjects in at least one of the study groups. Three samples with low-quality Ct values for over half of the analyzed genes were also removed from the analysis.

After data filtering, the dataset contained 5.58% of the missing values; therefore, a data imputation procedure was implemented. To minimize imputation bias, instead of calculating simple mean values, a multivariate linear regression approach was employed for data imputation.

The complete Ct dataset was analyzed using the delta Ct (dCt) method for relative quantification [85,86]. For each sample, normalization of the Ct values of the target gene was performed using the Ct values of the endogenous control gene (glyceraldehyde-3-phosphate dehydrogenase, *GAPDH*). This was achieved by subtracting the mean Ct value of *GAPDH* replicates from the mean Ct value of the target gene replicates to compute the delta Ct (dCt) values. Subsequently, the dCt values were converted to a linear, directly proportional form, using the 2^−dCt^ formula. Uniformity of the 2^−dCt^ data was evaluated using hierarchical clustering (Appendix A) and PCA (Appendix A). These assessments identified five samples as outliers, leading to their exclusion from further analysis. The remaining data demonstrated good homogeneity and were subsequently used for downstream analysis.

Differences in the expression of the studied genes were assessed through pairwise comparisons between the LEAD, AAA, and VV groups (i.e., LEAD vs. AAA, LEAD vs. VV, and AAA vs. VV). Fold-change values were calculated by dividing the mean 2^−dCt^ values of the first group in each comparison by the mean 2^−dCt^ values of the second group. Statistical analysis of gene expression differences was conducted following the methodology outlined in Section 4.6.

### 4.3. Plasma Protein Levels Data Analysis

This dataset contained information about the plasma concentrations of six proteins (ANGPT-1, ANGPT-2, TGF-alpha, TGF-beta 1, VEGF-A, and VEGF-C) in 40 patients with LEAD (LEAD group), 40 patients with AAA (AAA group), and 40 patients with VV (VV group), the same as for gene expression dataset. The analyzed proteins were quantified in plasma using commercially available ELISA kits (Appendix A).

A quality control procedure for this dataset included evaluation of the number of samples within and below the quantification range (Appendix A). In addition, data consistency was examined using hierarchical clustering (Appendix A) and PCA (Appendix A), which identified one outlier sample. Statistical analyses of the differences in plasma protein levels between the LEAD, AAA, and VV groups were performed using the same comparison framework as for the gene expression dataset (LEAD vs. AAA, LEAD vs. VV, and AAA vs. VV) according to the methodology detailed in Section 4.6.

### 4.4. RNA-Seq Dataset Analysis

RNA-seq dataset was previously obtained by our research team in the PBMC of eight patients with LEAD, seven patients with AAA, and seven patients with VV [41,42,43,82]. This dataset included the raw reads counts of 55,765 genes and was used to identify TFs potentially associated with dysregulations of genes selected as significant from the gene expression dataset analysis. Prior to the analysis, TFs regulating the selected genes were identified using the TFLink database, which provides extensive data on TF–target gene interactions across several model organisms [38]. For humans, TFLink contains information on 6,739,357 interactions involving 1606 TFs and 20,139 target genes.

The expression levels of the identified TFs were extracted from used RNA-seq dataset and differential expression analysis was performed using the DESeq2 1.42.1 package in R [39] to compare TFs expression between the disease groups (i.e., LEAD vs. AAA, LEAD vs. VV, and AAA vs. VV).

Data quality was assessed through multiple steps. TFs with a mean reads counts < 10 were excluded from the analysis. The homogeneity of the normalized data was evaluated using boxplots of Cook’s distances and PCA of the regularized log-transformed data (Appendix A), with no outliers detected. The quality of the differential expression analysis results was further examined using MA plots (Appendix A) and histograms of false discovery rate (FDR) values (Appendix A). Differentially expressed TFs were considered statistically significant if they met the following criteria: FDR < 0.05, |log_2_ fold change| > 1, and mean normalized count > 10.

### 4.5. miRNA-Seq Dataset Analysis

The miRNA-seq dataset used in this study comprises the expression of 2792 miRNA transcripts in the PBMC of 40 patients with LEAD, 28 patients with AAA, and 34 patients with VV [41,42,43,83]. This dataset was used in the present study to identify miRNAs potentially associated with differentially expressed genes selected from the gene expression dataset analysis, as well as TFs selected from RNA-seq dataset analysis.

First, the miRNet 2.0 database (https://www.mirnet.ca/) [40] was used to identify miRNAs predicted to interact with the selected genes and TFs. Experimentally validated interactions were obtained from the miRTarBase 9.0, TarBase 9.0, and miRecords resources, integrated with the miRNet 2.0 database.

Expression data for the identified miRNAs were extracted from this miRNA-seq dataset and subsequently a differential expression analysis was conducted using the DESeq2 1.42.1 package [39]. miRNA expression was compared between the LEAD and AAA groups, LEAD and VV groups, as well as between the AAA and VV groups. Consistent with the RNA-seq dataset analysis, miRNAs with a mean reads counts < 10 were filtered prior the analysis.

The homogeneity of the normalized data was assessed using boxplots of Cook’s distances and PCA of the regularized log-transformed data (Appendix A), confirming the absence of outliers. Quality control of the differential expression analysis was performed using MA plots (Appendix A) and histograms (Appendix A) of FDR values. Statistically significant miRNAs were defined as those meeting the following criteria: FDR < 0.05, |log_2_ fold change| > 0.3, and a mean normalized count > 10.

### 4.6. Statistical Analysis and Modeling

Statistical analysis was performed using the R 4.3.2 environment (https://www.r-project.org/).

Relationships between continuous variables were examined using correlation analysis, as well as univariate and multivariate linear regression. Correlation analysis was conducted using the Spearman rank correlation test, implemented in the Hmisc 5.1-1 package (https://cran.r-project.org/web/packages/Hmisc/index.html (accessed on 31 July 2025)). Linear regression models were developed utilizing the lm base function in R.

Univariate and multivariate logistic regression analyses, along with two statistical tests (Student’s *t*-test and Mann–Whitney U test), were employed to evaluate associations between continuous and categorical variables. The choice of the appropriate test was determined by assessing data normality using the two-sided Shapiro–Wilk test (shapiro.test function in R). When the distributions of the analyzed variables in both comparison groups were considered normal (*p* > 0.05, Shapiro–Wilk test), the parametric two-sided Student’s *t*-test was applied. Conversely, when the distribution in at least one of the groups was non-normal (*p* < 0.05, Shapiro–Wilk test), the non-parametric two-sided Mann–Whitney U test was used. Student’s *t*-test was conducted using the t.test function in R, while the Mann–Whitney U test was performed using the wilcox_test function in the coin 1.4-3 package in R [87]. Logistic regression models were constructed utilizing the base glm function in R.

Associations between categorical variables were evaluated using Fisher’s exact test, implemented in the fisher.test function in R.

To limit the risk of false positive results, *p* values derived from gene expression, plasma protein levels, TF and miRNA analyses, as well as from logistic regression, analysis of relationships with patients characteristics, as well as correlation and functional analyses, were adjusted using Benjamini–Hochberg method. Resulted FDR values below the threshold of 0.05 were considered as statistically significant.

Receiver operating characteristic (ROC) analysis was performed using the pROC 1.18.5 package in R [88]. Decision tree models were created using the C50 0.1.8 package in R (https://cran.r-project.org/web/packages/C50/index.html (accessed on 31 July 2025)).

## 5. Conclusions

The present study provides novel insights into the shared and distinct molecular mechanisms underlying the pathophysiology of LEAD, AAA, and VV. The identified gene and protein expression signatures may serve as potential biomarkers, enabling more accurate differentiation among these vascular conditions. This could contribute to reducing diagnostic overlap, minimizing delays in diagnosis, and improving treatment outcomes. However, further studies are required to validate these findings and establish their clinical applicability.

## Figures and Tables

**Figure 1 ijms-26-08786-f001:**
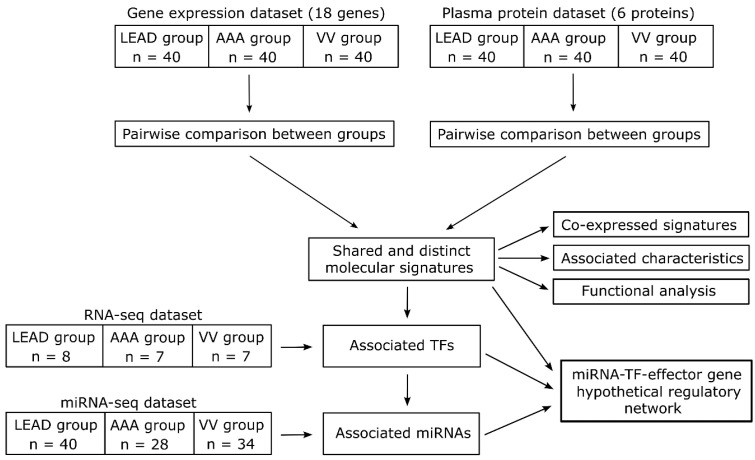
Summary of the study workflow. The study involves an analysis of gene expression and plasma protein datasets to identify shared and distinct molecular signatures between groups of patients with lower extremity artery disease (LEAD), abdominal aortic aneurysm (AAA), and varicose veins (VV). Co-expressed signatures, associations with demographic and clinical characteristics, and related biological processes were identified. Furthermore, transcription factors (TFs) and microRNAs (miRNAs) potentially regulating the obtained signatures were identified and integrated into a hypothetical miRNA-TF-effector gene regulatory network.

**Figure 2 ijms-26-08786-f002:**
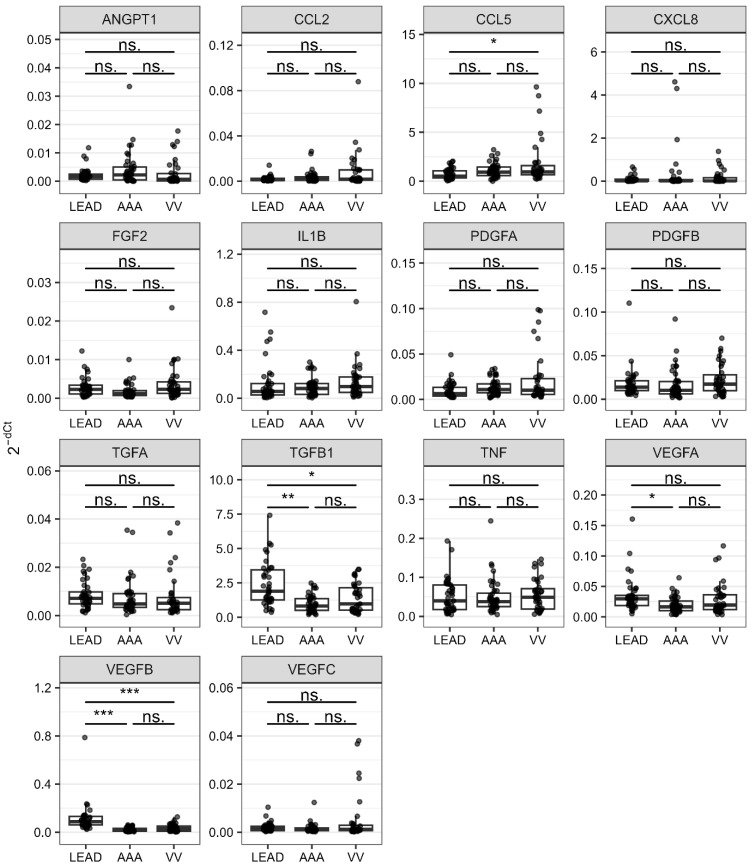
The distribution of 2^−dCt^ values calculated for 14 analyzed genes in the groups of patients with lower extremity artery disease (LEAD), abdominal aortic aneurysm (AAA), and varicose veins (VV). Whiskers reach the most distant point in the doubled interquartile range. Samples located outside the whiskers are marked as round points. Boxes range between 25% and 75% quartiles. Horizontal lines inside boxes mark the median values. Statistical significance was labeled as follows: ns.—not significant (FDR > 0.05), *—5 × 10^−2^ > FDR > 5 × 10^−4^, **—5 × 10^−4^ > FDR > 5 × 10^−6^, ***—FDR < 5 × 10^−6^. AAA—the group of patients with abdominal aortic aneurysm, LEAD—the group of patients with lower extremity artery disease, VV—the group of patients with varicose veins.

**Figure 3 ijms-26-08786-f003:**
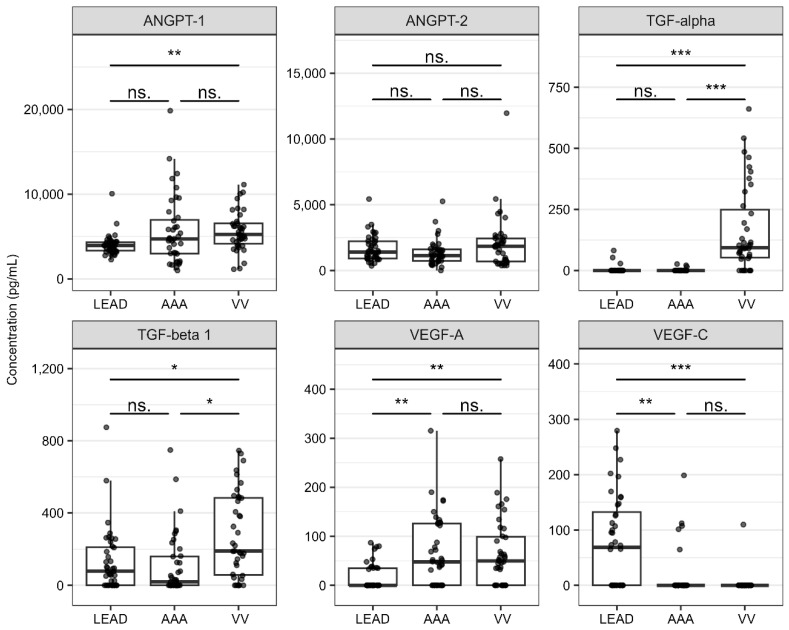
Distributions of plasma levels obtained for analyzed proteins in the groups of patients with lower extremity artery disease (LEAD), abdominal aortic aneurysm (AAA), and varicose veins (VV). Whiskers reach the most distant point in the doubled interquartile range, boxes range between 25% and 75% quartiles, and horizontal lines inside boxes mark the median value. Statistical significance (FDR) was labeled as follows: ns.—not significant (FDR > 0.05), *—5 × 10^−2^ > FDR > 5 × 10^−4^, **—5 × 10^−4^ > FDR > 5 × 10^−6^, ***—FDR < 5 × 10^−6^.

**Figure 4 ijms-26-08786-f004:**
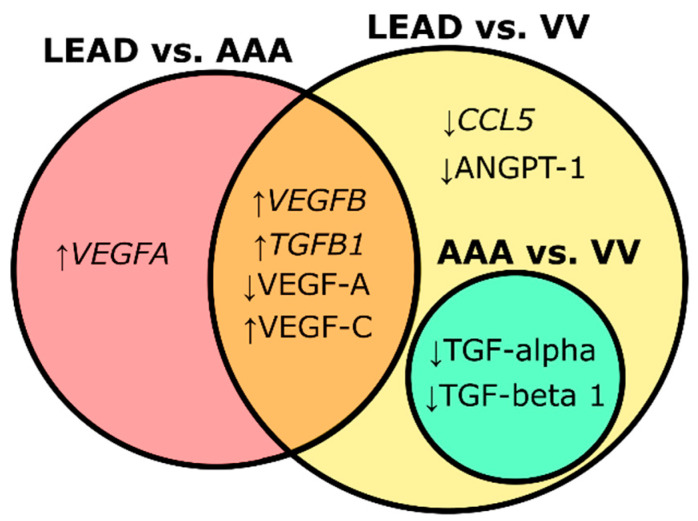
Shared and unique gene (in italic font) and protein (in normal font) signatures identified through pairwise comparisons among patient groups with lower extremity artery disease (LEAD), abdominal aortic aneurysm (AAA), and varicose veins (VV). Upward arrows denote increased expression levels, whereas downward arrows indicate decreased expression levels of the respective genes or proteins.

**Figure 5 ijms-26-08786-f005:**
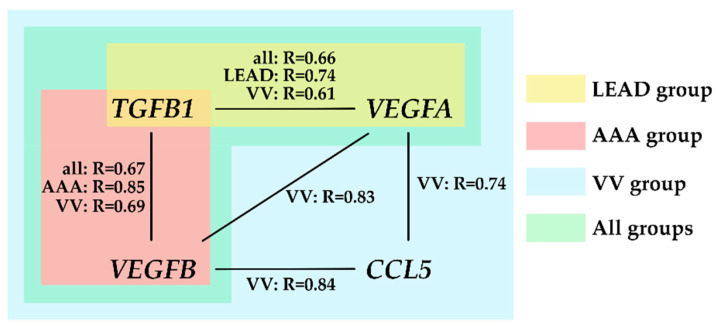
The strongest correlations (R > |0.6|) between the analyzed molecular signatures, identified within the groups of patients with lower extremity artery disease (LEAD group), abdominal aortic aneurysm (AAA group), and varicose veins (VV group), as well as across the entire cohort (All groups). R—Spearman correlation coefficient.

**Figure 6 ijms-26-08786-f006:**
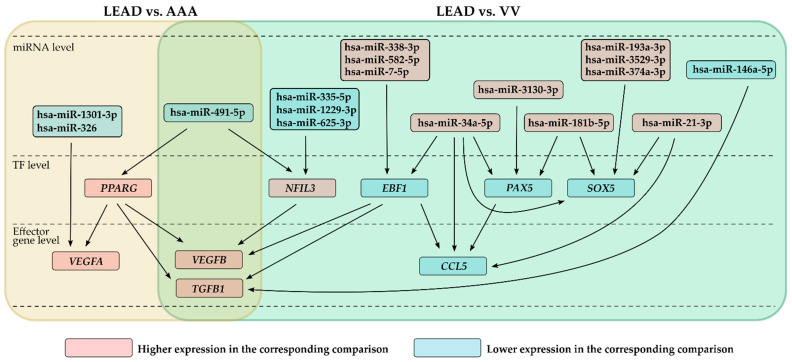
A hypothetical regulatory network depicting potential interactions among differentially expressed miRNAs, transcription factors (TFs), and effector genes was constructed based on comparative analyses between groups of patients with lower extremity artery disease (LEAD) and abdominal aortic aneurysm (AAA), as well as between groups of patients with lower extremity artery disease (LEAD) and varicose veins (VV).

**Figure 7 ijms-26-08786-f007:**
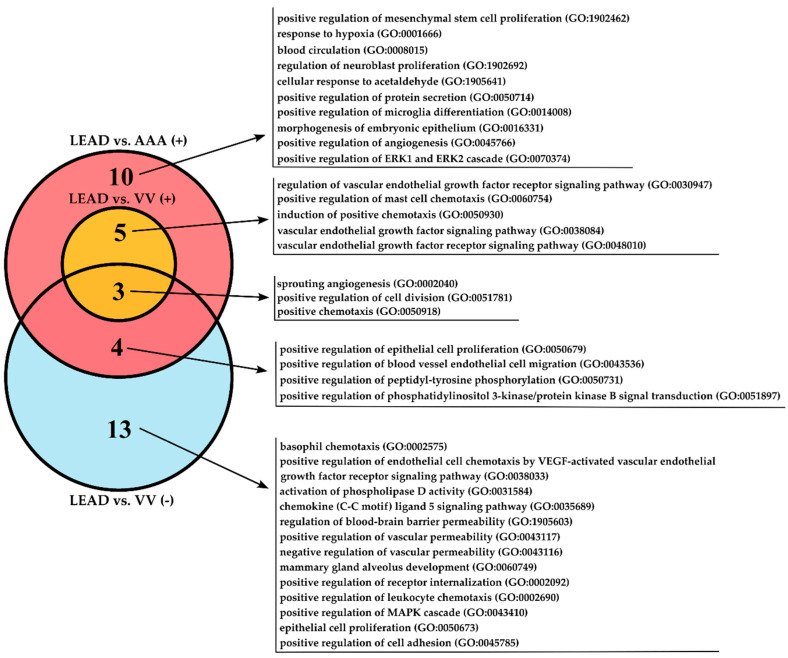
Gene Ontology Biological Process terms associated with genes and proteins signatures obtained as upregulated in LEAD vs. AAA comparison (LEAD vs. AAA (+)), upregulated from LEAD vs. VV comparison (LEAD vs. VV (+)), and downregulated from LEAD vs. VV comparison (LEAD vs. VV (−)). LEAD—group of patients with lower extremity artery disease, AAA—group of patients with abdominal aortic aneurysm, VV—group of patients with varicose veins.

**Table 1 ijms-26-08786-t001:** Comparison of the study groups.

Characteristic	LEAD Group (n = 40)	AAA Group (n = 40)	VV Group (n = 40)	*p* ^1^
Age	60.3 ± 7.56 (45–76)	59.2 ± 9.47 (45–80)	53.7 ± 8.00 (39–72)	2.058 × 10^−3^
Sex male/female	25 (62.5%)/15 (37.5%)	34 (85%)/6 (15%)	27 (67.5%)/13 (32.5%)	>0.05
Body mass index (BMI)	26.2 ± 3.06 (20.8–32.9)	26.8 ± 4.27 (19.5–35.1)	25.8 ± 3.40 (17.6–32.5)	>0.05
Smoking	35 (87.5%)	15 (37.5%)	0 (0%)	2.367 × 10^−17^
Hypertension	38 (95%)	7 (17.5%)	1 (2.5%)	1.151 × 10^−21^
LDL (mg/dL)	109.9 ± 21.04 (73–167)	108.9 ± 14.16 (79–151)	96.6 ± 14.23 (71–121)	1.513 × 10^−3^
HDL (mg/dL)	41.93 ± 2.90 (35–47)	40.70 ± 3.70 (31–46)	41.70 ± 3.52 (33–48)	>0.05
Cholesterol (mg/dL)	201.1 ± 11.2 (176–231)	206.1 ± 22.3 (143–302)	203.2 ± 18.2 (167–242)	>0.05
Creatinine (mg/dL)	0.87 ± 0.12 (0.67–1.09)	0.80 ± 0.16 (0.38–1.08)	0.63 ± 0.13 (0.34–0.89)	2.366 × 10^−10^
Urea (mg/dL)	33.0 ± 6.21 (21–56)	35.5 ± 5.71 (25–44)	31.2 ± 6.51 (21–45)	6.432 × 10^−3^
C-reactive protein (mg/L)	4.68 ± 1.40 (2.0–9.1)	4.34 ± 1.31 (1.1–6.8)	2.54 ± 1.06 (0.8–5.2)	1.779 × 10^−10^
Fibrinogen (mg/dL)	265.1 ± 60.3 (189–467)	177.5 ± 45.4 (121–316)	162.7 ± 33.5 (109–261)	8.834 × 10^−14^
Homocysteine (µmol/L)	6.92 ± 1.58 (4.99–12.7)	8.00 ± 2.04 (3.56–13.8)	6.44 ± 1.35 (3.89–8.9)	2.772 × 10^−4^

^1^ For continuous-type variables (age, BMI, and biochemical blood parameters), statistical significance was calculated using the two-sided Student’s *t*-test or the two-sided Mann–Whitney U test (depending on the normality of the data). For categorical variables (sex, smoking, and hypertension), two-sided Fisher’s exact test was used. Continuous variables are presented as mean ± SD, and range in brackets. Categorical variables are presented as counts and percentages. LDL—low-density lipoprotein, HDL—high-density lipoprotein.

**Table 2 ijms-26-08786-t002:** Differentially expressed genes resulted with statistical significance from comparisons performed between the LEAD, AAA, and VV groups.

Comparison	Gene Symbol	Gene Name	Differential Expression		ROC	Univariate Logistic Regression	
			Fold Change	FDR	ROC-AUC	OR	FDR
LEAD vs. AAA	*TGFB1*	Transforming growth factor beta 1	2.494	2.704 × 10^−5^	0.806	9.997	1.297 × 10^−3^
	*VEGFA*	Vascular endothelial growth factor A	1.858	1.490 × 10^−3^	0.739	4.432	1.995 × 10^−2^
	*VEGFB*	Vascular endothelial growth factor B	5.057	1.099 × 10^−10^	0.953	526.126	4.312 × 10^−4^
LEAD vs. VV	*CCL5*	C-C motif chemokine ligand 5	0.405	3.143 × 10^−2^	0.686	0.351	0.127
	*TGFB1*	Transforming growth factor beta 1	1.716	3.143 × 10^−2^	0.695	3.089	4.201 × 10^−2^
	*VEGFB*	Vascular endothelial growth factor B	3.248	5.914 × 10^−7^	0.876	26.394	5.374 × 10^−4^

The provided gene symbols and names are in accordance with the actual nomenclature in the HUGO Gene Nomenclature Committee (HGNC) (https://www.genenames.org/). FDR—false discovery rate (*p* values adjusted by Benjamini–Hochberg method), OR—odds ratio, ROC-AUC—area under receiver operating characteristics curve. FDR values considered statistically significant (FDR < 0.05) are presented in a scientific format (exponential notation).

**Table 3 ijms-26-08786-t003:** Differences in plasma protein levels between the LEAD and AAA groups.

Comparison	Protein Symbol	Protein Name	Mean Concentration (pg/mL) ± SD		FDR	ROC-AUC	Univariate Logistic Regression	
			LEAD	AAA			OR	FDR
LEAD vs. AAA	VEGF-A	Vascular endothelial growth factor A	15.92 ± 27.01	66.40 ± 71.97	4.969 × 10^−4^	0.726	0.978	2.452 × 10^−4^
	VEGF-C	Vascular endothelial growth factor C	76.28 ± 83.13	14.60 ± 42.20	3.637 × 10^−4^	0.719	1.016	2.452 × 10^−4^
LEAD vs. VV	ANGPT-1	Angiopoietin-1	4021.43 ± 1250.10	5592.54 ± 2309.60	1.069 × 10^−4^	0.764	0.999	3.085 × 10^−3^
	TGF-alpha	Protransforming growth factor alpha	4.10 ± 15.79	165.05 ± 175.22	5.000 × 10^−10^	0.883	0.953	2.950 × 10^−4^
	TGF-beta 1	Transforming growth factor beta-1 proprotein	132.38 ± 174.67	277.53 ± 236.20	6.833 × 10^−3^	0.679	0.996	6.571 × 10^−3^
	VEGF-A	Vascular endothelial growth factor A	15.92 ± 27.01	64.36 ± 65.08	1.111 × 10^−4^	0.742	0.975	1.871 × 10^−3^
	VEGF-C	Vascular endothelial growth factor C	76.28 ± 83.13	2.81 ± 17.56	1.529 × 10^−6^	0.763	1.034	3.085 × 10^−3^
AAA vs. VV	TGF-alpha	Protransforming growth factor alpha	1.60 ± 5.83	165.05 ± 175.22	2.417 × 10^−10^	0.890	0.929	4.604 × 10^−3^
	TGF-beta 1	Transforming growth factor beta-1 proprotein	106.29 ± 171.70	277.53 ± 236.20	9.934 × 10^−4^	0.730	0.996	4.604 × 10^−3^

The protein names were in accordance with the UniProt database (release 2024_01, https://www.uniprot.org/). FDR—statistical significance calculated by two-sided Mann–Whitney U test and adjusted by Benjamini–Hochberg correction rate; ROC-AUC—area under receiver operating characteristics curve. SD—standard deviation.

**Table 4 ijms-26-08786-t004:** Results of differential expression analysis of transcription factors.

Comparison	TF Symbol	TF Name	Differential Expression		ROC	Selected Associated Genes
			Fold Change	FDR	ROC-AUC	
LEAD vs. AAA	*PPARG*	peroxisome proliferator activated receptor gamma	4.034	3.247 × 10^−2^	0.857	↑ *VEGFA*, ↑ *VEGFB*, ↑ *TGFB1*
LEAD vs. VV	*EBF1*	EBF transcription factor 1	0.392	1.862 × 10^−2^	0.929	↑ *VEGFB*, ↑ *TGFB1,* ↓ *CCL5*
	*NFIL3*	nuclear factor, interleukin 3 regulated	2.119	1.306 × 10^−2^	0.929	↑ *VEGFB*
	*PAX5*	paired box 5	0.346	8.957 × 10^−3^	0.875	↑ *TGFB1*, ↓ *CCL5*
	*SOX5*	SRY-box transcription factor 5	0.227	3.609 × 10^−4^	1.000	↑ *TGFB1*

Provided gene symbols and gene names are in accordance with actual nomenclature in HUGO Gene Nomenclature Committee (HGNC) (https://www.genenames.org/). FDR—false discovery rate (*p* values adjusted by Benjamini–Hochberg method), ROC-AUC—area under receiver operating characteristics curve. Upward arrows denote increased expression levels, whereas downward arrows indicate decreased expression levels of the respective gene.

**Table 5 ijms-26-08786-t005:** Results of differential expression analysis of selected miRNAs.

Comparison	miRNA	Differential Expression		ROC	Associated Genes or TFs
		Fold Change	FDR	ROC-AUC	
LEAD vs. AAA	hsa-miR-1301-3p	0.741	1.398 × 10^−4^	0.815	↑ *VEGFA*
	hsa-miR-326	0.767	1.278 × 10^−2^	0.739	↑ *VEGFA*
	hsa-miR-491-5p	0.809	4.086 × 10^−2^	0.668	↑ *PPARG*
LEAD vs. VV	hsa-miR-181b-5p	1.441	8.738 × 10^−4^	0.752	↓ *PAX5*,↓ *SOX5*
	hsa-miR-3130-3p	1.359	1.023 × 10^−2^	0.779	↓ *PAX5*
	hsa-miR-193a-3p	1.368	1.023 × 10^−2^	0.741	↓ *SOX5*
	hsa-miR-1229-3p	0.621	1.398 × 10^−2^	0.692	↑ *NFIL3*
	hsa-miR-338-3p	1.234	1.398 × 10^−2^	0.685	↓ *EBF1*
	hsa-miR-7-5p	1.255	1.945 × 10^−2^	0.688	↓ *EBF1*
	hsa-miR-146a-5p	0.811	2.289 × 10^−2^	0.735	↑ *TGFB1*
	hsa-miR-3529-3p	1.240	2.842 × 10^−2^	0.672	↓ *SOX5*
	hsa-miR-491-5p	0.795	2.915 × 10^−2^	0.657	↑ *NFIL3*
	hsa-miR-625-3p	0.727	3.691 × 10^−2^	0.672	↑ *NFIL3*
	hsa-miR-335-5p	0.761	3.691 × 10^−2^	0.691	↑ *NFIL3*
	hsa-miR-374a-3p	1.266	3.691 × 10^−2^	0.657	↓ *SOX5*
	hsa-miR-582-5p	1.318	4.509 × 10^−2^	0.667	↓ *EBF1*
	hsa-miR-34a-5p	1.279	4.890 × 10^−2^	0.726	↓ *CCL5*, ↓ *EBF1* ↓ *PAX5*, ↓ *SOX5*
	hsa-miR-21-3p	1.286	4.960 × 10^−2^	0.682	↓ *SOX5*, ↓ *CCL5*

Provided miRNA names are in accordance with actual nomenclature in miRBase database (https://www.mirbase.org/). FDR—false discovery rate (*p* values adjusted by Benjamini–Hochberg method), ROC-AUC—area under receiver operating characteristics curve. Upward arrows denote increased expression levels, whereas downward arrows indicate decreased expression levels of the respective gene.

## Data Availability

The data used for this study are openly available in FigShare repository. The gene expression and protein plasma levels dataset for AAA group can be found at https://doi.org/10.6084/m9.figshare.23791485.v1. The gene expression and protein plasma levels dataset for VV group is available at https://doi.org/10.6084/m9.figshare.26065462.v1. The link for gene expression and protein plasma levels dataset for LEAD group will be provided upon acceptance. The miRNA-seq dataset can be find at https://doi.org/10.6084/m9.figshare.19446851.v2. The RNA-seq dataset can be find at https://doi.org/10.6084/m9.figshare.14252897.v1.

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
