# Peer review of "Molecular Signatures Related to Inflammation and Angiogenesis in Patients with Lower Extremity Artery Disease, Abdominal Aortic Aneurysm, and Varicose Veins: Shared and Distinct Pathways"

_ijms, 2025, doi:10.3390/ijms26188786_

Round 1

Reviewer 1 Report

Comments and Suggestions for Authors

Dear Authors,

Major Comments

  • The scope of multiple testing correction (e.g., FDR) is unclear. The analyses subject to correction and the method used should be explicitly stated.
  • The same analysis results are repeated in the text and should be simplified, especially in sections 2.2–2.3.
  • The inverse correlation between VEGFA and TGFB1 gene expression and plasma levels is not sufficiently explained. Comparison with previous studies is needed.
  • In the TF analysis, the assumption of “upregulation” is made, but the possibility of inhibitory effects should also be acknowledged.
  • The miRNA-TF-gene network is based on predictions; therefore, expressions such as “may suggest” should be used.
  • In Figure 6, the subject groups should be clearly indicated to avoid confusion.
  • Limitations regarding VEGFA isoforms and gene expression should be discussed.
  • The limitations regarding clinical applications and diagnostic accuracy should be clearly stated.

Minor Comments

  • Long sentences should be revised for clarity, with improved punctuation and structure for readability.
  • Abbreviations (e.g., ROC-AUC, OR, FDR) should be explained at first mention.

Author Response

Dear Reviewer,

We sincerely thank you for your valuable and insightful comments on our manuscript entitled “Molecular Signatures related to Inflammation and Angiogenesis in Patients with Lower Extremity Artery Disease, Abdominal Aortic Aneurysm, and Varicose Veins: Shared and Distinct Pathways” (submission ID ijms-3825103). We have carefully read all the suggestions and have made the necessary revisions accordingly. We found all of the comments to be constructive and highly beneficial in improving the quality and clarity of our paper. We are submitting a revised version of the manuscript, in which all changes have been highlighted in red font for easier reference. Below, we provide a point-by-point response to each of the Reviewer comments:

Response to the Reviewer 1

Major Comments

Comment:

The scope of multiple testing correction (e.g., FDR) is unclear. The analyses subject to correction and the method used should be explicitly stated.

Response:

We appreciate this insightful comment, which indeed required further clarification. Our study involved multiple statistical analyses, and we agree that the sections in which correction for multiple testing was applied were not sufficiently explicit in the manuscript. To address this, we revised the statement

“Results with FDR < 0.05 (p values adjusted for multiple testing using the Benjamini–Hochberg method) were considered statistically significant, unless otherwise indicated.” (lines 799-801 in the original version of the manuscript)

to the following:

“To limit the risk of false positive results, p values derived from gene expression, plasma protein levels, TF and miRNA analyses, as well as from logistic regression, analysis of relationships with patients characteristics, as well as correlation and functional analyses, were adjusted using Benjamini-Hochberg method. Resulted FDR values below the threshold of 0.05 were considered as statistically significant.” (lines 841-845).

This revision explicitly specifies the analyses in which FDR correction was applied. In addition, we clarified throughout the manuscript where adjusted p values were reported instead of raw p values (lines 173-175, 198, 256, and 270).

Comment:

The same analysis results are repeated in the text and should be simplified, especially in sections 2.2–2.3.

Response:

Thank you for this suggestion. We agree that these sections are quite similar, as contain results obtained with the same methodology. However, these sections include results obtained using different datasets (expression levels of 18 genes in PBMC vs. plasma levels of 6 proteins), different data homogeneity assessment outcomes, different numbers of statistically significant genes/proteins selected from performed comparisons, and different results of logistic regression and decision tree analyses. Due to this numerous and often subtle differences, we intentionally presented these sections in a detailed manner to ensure precision, clarity, and unambiguous interpretation of the information provided. We therefore kindly ask for the reviewer’s understanding if we retain the text in its current form, and we sincerely hope this explanation clarifies our reasoning.

Comment:

The inverse correlation between VEGFA and TGFB1 gene expression and plasma levels is not sufficiently explained. Comparison with previous studies is needed.

Response:

We agree with the Reviewer that these findings need greater emphasis. We have expanded the Discussion section to address potential mechanisms underlying the observed discrepancies between PBMC gene expression and plasma concentrations of TGF-beta 1 and VEGF-A. In this revised section, we provide additional interpretation supported by relevant literature and our own data (lines 635-668).

Comment:

In the TF analysis, the assumption of “upregulation” is made, but the possibility of inhibitory effects should also be acknowledged.

Response:

We appreciate this valuable comment. In our initial analysis, we focused primarily on the role of TFs as activators of gene expression; however, TFs may also act as repressors of DNA transcription. In response to this point, we reanalyzed our TF results and included all statistically significant TFs associated with the genes identified in the corresponding comparisons. As a result, the list of selected TFs was expanded to include SOX5 and EBF1. The main text (lines 387-392, 399, 403-405), Table 4, Table S11, as well as Figures 6, S28, and S29 have been updated to reflect these revisions.

In addition, we repeated the miRNA analysis to incorporate miRNAs potentially regulating the two newly added TFs. Consequently, Section 2.7, together with Figures 5, S30–S34 and Tables 5, S12, and S13, were revised to present the updated findings.

The Discussion section was also expanded to integrate these results and provide a more comprehensive interpretation (lines 589, 597-607).

Comment:

The miRNA-TF-gene network is based on predictions; therefore, expressions such as “may suggest” should be used.

Response:

We thank the Reviewer for this comment. We agree that certain statements and conclusions in the manuscript would benefit from being expressed with less definitive language and a more cautious, predictive tone. We have revised the text and introduced appropriate changes to address this issue (lines 406, 441, 444, 446, 448, 458-459, 575, 599, 624).

Comment:

In Figure 6, the subject groups should be clearly indicated to avoid confusion.

Response:

We agree with the reviewer that Figure 6 could be improved for better visual separation of the results corresponding to the different comparisons. In the original version, the results from the LEAD vs. AAA and LEAD vs. VV comparisons were distinguished by color, with the overlapping regions remaining visible. To enhance readability, we have implemented several modifications:

  • the frames surrounding the results were thickened and rendered in a darker shade of their original color,
  • the font size for the labels “LEAD vs. AAA” and “LEAD vs. VV” was increased from 11 pt to 13 pt,
  • the corners of the fields were more rounded to improve visual separation.

We trust that these refinements substantially increase clarity and address the reviewer’s concerns.

Comment:

Limitations regarding VEGFA isoforms and gene expression should be discussed.

Response:

We appreciate the Reviewer’s thoughtful feedback. In the original version of our manuscript (lines 531–543), we discussed the impact of transcript variability in VEGF-A on its biological functions, which may differ depending on the isoform. We agree that this aspect needs a more comprehensive description. Therefore, we have expanded the section on study limitations to explicitly address this issue (lines 674–679).

Comment:

The limitations regarding clinical applications and diagnostic accuracy should be clearly stated.

Response:

We thank the Reviewer for highlighting these important point, which indeed needs our attention. As our work regards results with potential clinical applicability, we agree that limitations in this aspect should be stated. Therefore, we extended the list of our study limitations to address limitations regarding clinical applications and diagnostic accuracy (lines 684-693).

Minor Comments

Comment:

Long sentences should be revised for clarity, with improved punctuation and structure for readability.

Response:

We agree with the Reviewer’s suggestion and have revised long sentences throughout the manuscript to improve clarity and readability (lines 63, 66-68, 78, 100-102, 239, 389, 425, 434, 444, 551)

Comment:

Abbreviations (e.g., ROC-AUC, OR, FDR) should be explained at first mention.

Response:

The manuscript has been revised to ensure that all abbreviations are defined at their first occurrence.

Reviewer 2 Report

Comments and Suggestions for Authors

The paper by Zalewski and collaborators identified distinct and shared gene and protein molecular

signatures of inflammation and angiogenesis among patients with lower extremity artery disease (LEAD), abdominal aortic aneurysm (AAA), and varicose veins (VV).  40 patients for each disease were enrolled and sampled for blood and PBMC. The study identified two factors that exhibited opposing trends in the PBMC and plasma samples. Specifically, an increased expression of VEGFA was observed between groups LEAD and AAA in PBMC, whereas plasma levels of VEGF-A were lower. A similar pattern was detected for TGFB1, with elevated expression in PBMC and lowered plasma levels in patients with LEAD compared to those with VV

Additionally by integrating RNA-seq and miRNA-seq data, the authors constructed miRNA/transcription factor/gene regulatory networks that may be involved in the dysregulation of inflammation and angiogenesis in these diseases.

The paper, which opens to further research to definitely focus on diagnostic and therapeutic targets for the vascular disease, is appropriately conduced and analysed and well written. The presented results sound and the authors made an accurate analysis of the study limitations.

Minor comment: 

There is no mention about the influence of sex on the findings. Please specify the statistical analysis result about this parameter.

Author Response

Dear Reviewer,

We sincerely thank you for your kind opinion on our manuscript entitled “Molecular Signatures related to Inflammation and Angiogenesis in Patients with Lower Extremity Artery Disease, Abdominal Aortic Aneurysm, and Varicose Veins: Shared and Distinct Pathways” (submission ID ijms-3825103). We have carefully read your comments and have made the necessary revisions accordingly. We found these comments constructive and valuable in improving the rigor and clarity of our paper. We are submitting a revised version of the manuscript, in which all changes have been highlighted in red font for easier reference. Below, we provide a point-by-point response to the Reviewer’s comments:

Response to the Reviewer 1

Comment:

The paper by Zalewski and collaborators identified distinct and shared gene and protein molecular signatures of inflammation and angiogenesis among patients with lower extremity artery disease (LEAD), abdominal aortic aneurysm (AAA), and varicose veins (VV).  40 patients for each disease were enrolled and sampled for blood and PBMC. The study identified two factors that exhibited opposing trends in the PBMC and plasma samples. Specifically, an increased expression of VEGFA was observed between groups LEAD and AAA in PBMC, whereas plasma levels of VEGF-A were lower. A similar pattern was detected for TGFB1, with elevated expression in PBMC and lowered plasma levels in patients with LEAD compared to those with VV

Additionally by integrating RNA-seq and miRNA-seq data, the authors constructed miRNA/transcription factor/gene regulatory networks that may be involved in the dysregulation of inflammation and angiogenesis in these diseases.

The paper, which opens to further research to definitely focus on diagnostic and therapeutic targets for the vascular disease, is appropriately conduced and analysed and well written. The presented results sound and the authors made an accurate analysis of the study limitations.

Response:

Once again, we greatly appreciate your constructive feedback.

Minor comment:

There is no mention about the influence of sex on the findings. Please specify the statistical analysis result about this parameter.

Response:

We consider this comment highly important for ensuring the comprehensiveness and rigor of presented study. We fully agree with the Reviewer that sex is an important factor that can significantly impact our results, as the epidemiological risk, clinical image, and prognosis of the studied diseases differs in men and women. To address this potential impact, we explored relationships between identified statistically significant gene and protein signatures (CCL5, TGFB1, VEGFA, VEGFB, ANGPT-1, TGF-alpha, TGF-beta 1, VEGF-A, and VEGF-C) and sex, among other categorical characteristics of the study groups (please refer to the section 2.4). Using a statistical tests suitable for data distribution (Mann-Whitney U test or Student’s t-test), no statistically significant (using Benjamini-Hochberg FDR < 0.05 threshold) differences in these factors were found between men and women. Resulted FDR values are presented in Table S8. Originally, the analysis was performed using all studied samples (not separately by the particular studied diseases), as sex and other categorical variables were originally considered as unbalanced across the studied diseases. However, in light of the Reviewer’s comment, we examined the sex variable in greater detail and subsequently decided to perform an additional analysis separately for each disease group, as no statistically significant differences in sex distributions were found between disease groups (refer to Table 1). This disease-specific analysis revealed no statistically significant differences in the expression levels of gene and protein signatures between men and women within the AAA, LEAD, and VV groups when applying the Benjamini-Hochberg FDR threshold of <0.05. Obtained new FDR values were added to Table S8. These results may suggest that analyzed factors are sex-independent; however, further studies on larger cohorts should be perform to validate this conclusion.

We thank the reviewer for highlighting these important point, which will receive greater attention in our future research.